# Infection-Related Cryoglobulinemic Glomerulonephritis with Serum Anti-Factor B Antibodies Identified and Staining for NAPlr/Plasmin Activity Due to Infective Endocarditis

**DOI:** 10.3390/ijms24119369

**Published:** 2023-05-27

**Authors:** Takumi Toishi, Takashi Oda, Atsuro Hamano, Shinnosuke Sugihara, Tomohiko Inoue, Atsuro Kawaji, Kanako Nagaoka, Masatoshi Matsunami, Junko Fukuda, Mamiko Ohara, Tomo Suzuki

**Affiliations:** 1Department of Nephrology, Kameda Medical Center, 929 Higashi-cho, Kamogawa 296-8602, Chiba, Japan; toishi.takumi@kameda.jp (T.T.);; 2Department of Nephrology and Blood Purification, Tokyo Medical University Hachioji Medical Center, 1163, Tate-machi, Hachioji 193-0998, Tokyo, Japan; takashio@tokyo-med.ac.jp

**Keywords:** cryoglobulinemic glomerulonephritis, infection-related glomerulonephritis, factor B, nephritis-associated plasmin receptor (NAPlr), infective endocarditis

## Abstract

In this rare case of infection-related cryoglobulinemic glomerulonephritis with infective endocarditis, a 78-year-old male presented with an acute onset of fever and rapidly progressive glomerulonephritis. His blood culture results were positive for *Cutibacterium modestum*, and transesophageal echocardiography showed vegetation. He was diagnosed with endocarditis. His serum immunoglobulin M, IgM-cryoglobulin, and proteinase-3-anti-neutrophil cytoplasmic antibody levels were elevated, and his serum complement 3 (C3) and C4 levels were decreased. Renal biopsy results showed endocapillary proliferation, mesangial cell proliferation, and no necrotizing lesions on light microscopy, with strong positive staining for IgM, C3, and C1q in the capillary wall. Electron microscopy showed deposits in the mesangial area in the form of fibrous structures without any humps. Histological examination confirmed a diagnosis of cryoglobulinemic glomerulonephritis. Further examination showed the presence of serum anti-factor B antibodies and positive staining for nephritis-associated plasmin receptor and plasmin activity in the glomeruli, suggesting infective endocarditis-induced cryoglobulinemic glomerulonephritis.

## 1. Introduction

Bacterial infection occasionally manifests as a complication of glomerulonephritis (GN), also referred to as infection-related glomerulonephritis (IRGN). IRGN serves as a collective term for glomerulonephritis resulting from post-infectious or ongoing infections, which gained widespread recognition following a report by Nasr et al. [1]. The typical manifestation of IRGN is an acute nephritic syndrome characterized by low serum levels of complement 3 (C3) [2]. Kidney biopsy utilizing light microscopy (LM) shows endocapillary proliferative glomerulonephritis, with electron microscopy (EM) often indicating a subepithelial deposit referred to as a hump. A diagnosis of IRGN can be challenging owing to the complexity and ongoing nature of infections in adults; however, recent advancements in diagnostic tools, such as nephritis-associated plasmin receptor (NAPlr) and anti-factor B (FB) antibodies, have facilitated the accurate diagnosis of IRGN. NAPlr is the same molecule as streptococcal glyceraldehyde-3-phosphate dehydrogenase (GAPDH) and serves as a histological marker of IRGN, as its presence strongly suggests a prior or ongoing infection. Glomerular positive staining for NAPlr and plasmin activity have been reported in many cases of IRGN [3]. Serum anti-FB antibodies, which activate alternative pathways during infections, are particularly useful for diagnosing IRGN in cases of GN with low serum C3 [4].

Cryoglobulins are pathological cold-sensitive antibodies that often present with multiple symptoms. Cryoglobulinemia has been classified into three types based on its diverse etiologies, including hematological, infectious, and autoimmune diseases [5]. Cryoglobulinemic glomerulonephritis shows irregular capillary wall subendothelial, mesangial, or globular intracapillary deposits with fibrillary or microtubular substructures [6]. Many cases of cryoglobulinemia are attributed to hepatitis C, with few reports of cryoglobulinemic glomerulonephritis associated with bacterial infections. 

Herein, we present a rare case of cryoglobulinemic glomerulonephritis resulting from infective endocarditis (IE). Kidney biopsy showed endocapillary proliferative glomerulonephritis with immunoglobulin M (IgM) deposition and fibrillary structures among the deposits. The association between diagnosing infection using glomerular NAPlr staining and anti-FB antibodies was also investigated.

## 2. Case Report 

A 78-year-old Japanese male, who developed an acute onset of fever, presented to the emergency department one week after symptom onset. His medical history included severe aortic stenosis following aortic valve replacement. His current medications included bisoprolol fumarate, olmesartan medoxomil, and rosuvastatin. 

Physical examination findings were as follows: a composed demeanor; blood pressure, 136/70 mmHg; pulse, 68 beats/min; respiratory rate, 12 breaths/min; oxygen saturation, 97% on room air; and body temperature, 40.0 °C. Upon head and neck examination, no enlargement of the cervical lymph nodes, dental decay, or petechiae in the conjunctivae or oral cavity was observed. On cardiovascular examination, a Levine II/VI apical systolic murmur was confirmed. No evidence of arthralgia or rashes, including Janeway lesions or Osler nodes, was noted.

Laboratory data were obtained on admission. Urinalysis showed hematuria and slight proteinuria (50–99 red blood cells per high-power field; spot urine protein concentration, 0.3 g/g Cr; and a white blood cell count in the range of 5–9 per high-power field without abnormalities in casts). His complete blood count was within normal limits (white blood cell count, 7500/μL; neutrophil count, 78.6%). Biochemical examination indicated a serum creatinine (Cr) and estimated glomerular filtration rate of 1.75 mg/dL and 30.12 mL/min/1.73 m^2^, respectively, and a C-reactive protein level of 2.33 mg/dL. Computed tomography (CT) scans showed bilateral kidney enlargement and left thalamic hemorrhage.

Upon further serological examination, a decrease in C3 and C4 complement levels was observed (59 mg/dL and 7.1 mg/dL, respectively). The IgM level was elevated (259 mg/dL), whereas other immunoglobulin levels were within the normal range. A blood test also indicated an elevated proteinase-3-anti-neutrophil cytoplasmic antibody (PR3-ANCA) level of 140 U/L. Cryoglobulin examination showed the presence of polyclonal IgM. The rheumatoid factor level was within normal limits (5 IU/mL). The blood culture was positive for *Cutibacterium modestum*. Furthermore, he developed bilateral conjunctival petechiae on hospital day five. Transesophageal echocardiography showed the presence of a 5 mm vegetation at the aortic valve. Based on these findings, a diagnosis of IE due to *Cutibacterium modestum* was established, and treatment with aminobenzylpenicillin was initiated. 

A kidney biopsy was performed. Of the 13 glomeruli obtained, one exhibited global sclerosis. LM showed endocapillary and mesangial cell proliferation without crescent formation, necrotizing lesions, or endarteritis (Figure 1). Additionally, there was infiltration of inflammatory cells into the interstitial parenchyma. Immunofluorescence staining (IF) indicated the presence of IgM (++), C3(++), and C1q (+) in the capillary wall (Figure 1B–F), with C3 also observed in the mesangial area. EM showed high-density electron deposits in the mesangial area without hump formation and macrophage infiltration (Figure 1G). At high-power magnification, these deposits appeared as fibrous structures (Figure 1H). The histological diagnosis was cryoglobulinemic glomerulonephritis. There was no indication of ANCA-associated vasculitis complications, given that there were no manifestations of necrotizing lesions or arteritis.

Further investigation of the kidney biopsy and serum samples showed an association between glomerulonephritis and IE. Firstly, we performed IF for NAPlr and in situ zymography for plasmin activity on serial sections of fresh frozen renal biopsy tissue. As for NAPlr, direct IF using fluorescein isothiocyanate (FITC)-conjugated polyclonal rabbit anti-NAPlr antibody was performed, while plasmin activity in the serial section was evaluated by in situ zymography using plasmin-sensitive synthetic peptide: p-toluenesulfonyl-L-lysine α-naphthyl ester (Torii Pharmaceutical Co., Ltd., Tokyo, Japan) as described previously [7]. Both NAPlr and plasmin activity were positively stained in the segmental areas of the glomeruli in a similar pattern (Figure 2A,B). The serum anti-FB antibodies level, assessed using an FB-coated ELISA plate, was 1.880 UA/mL, indicating a strong positive result (reference range, <0.237 UA/mL). This suggests that cryoglobulinemic glomerulonephritis was induced through an infection-related process, specifically IE. 

Surgery was considered due to prosthetic valve endocarditis; however, our patient was not considered a suitable surgery because of the concurrent left thalamic hemorrhage. During treatment, he exhibited signs of malnutrition and disuse syndrome. He was unable to return home and was transferred to another hospital.

## 3. Discussion

We report a rare case of cryoglobulinemic glomerulonephritis attributable to IE in which the diagnosis of kidney disease was substantiated using a histological examination of the kidney tissue (NAPlr and plasmin staining) and serum markers, which showed a particularly robust positive level of serum anti-FB antibodies. 

Cryoglobulinemic glomerulonephritis commonly stems from hepatitis C infection, with few other infectious causes having been reported. Several studies have reported the association between cryoglobulinemia and Bartonella infections; however, Arani et al. [8,9] reported proliferative glomerulonephritis with deposits of polyclonal IgG. One study reported that hepatitis E infection-induced glomerulonephritis was accompanied by monoclonal IgG cryoglobulinemia [10]. Further, one case series documented staphylococcal IRGN with cryoglobulinemic features [11]. In that report, five cases exhibited IgA-dominant glomerulonephritis in the presence of cryoglobulins. These findings indicated that all cases were characterized by proliferative and exudative glomerulonephritis accompanied by hyaline pseudothrombi.

The kidney biopsy findings concerning our patient indicated proliferative glomerulonephritis with deposits of polyclonal IgM in the capillary walls and fibrous structures, thereby corroborating the presence of type IgM cryoglobulinemia due to IE and its associated glomerulonephritis. In addition, we found positive glomerular staining for NAPlr/plasmin activity and the presence of serum anti-FB antibodies.

NAPlr, a recently established diagnostic biomarker for IRGN, plays a dual role as both a plasmin receptor and an activator of the alternative complement pathway. The presence of NAPlr/bacterial GAPDH in circulation, which is released in response to bacterial infection, results in its trapping by the glomeruli and binding with plasmin, which initiates the degradation of extracellular matrix proteins, leading to the accumulation of neutrophils and macrophages, and thus, endocapillary glomerular inflammation [12]. Additionally, NAPlr directly activates the alternative complement pathway through the conversion of complement component C3 to C3b [13]. Therefore, staining for NAPlr/plasmin activity provides a diagnostic tool for IRGN. While the presence of NAPlr/plasmin activity has been documented in several variants of glomerulonephritis, including IgA-dominant and C3-dominant glomerulonephritis, no such positivity has been reported in cases of cryoglobulinemic glomerulonephritis.

Anti-FB antibodies are autoimmune agents that target FB. Chauvet et al. reported a correlation between the presence of anti-FB antibodies, C3, and IRGN [4]. FB is a component of the alternative pathway C3 convertase, and the presence of serum anti-FB antibodies was found to activate the alternative pathway. A correlation between serum anti-FB antibody titers and hypocomplementemia severity was also identified. The normalization of C3 levels was associated with a decline in anti-FB antibodies. Furthermore, an elevation in anti-FB antibodies was noted during the period of infection, which normalized following the resolution of the infection. We observed elevated levels of anti-FB antibodies and deposition of NAPlr with elevated plasmin activity in the glomeruli, consistent with IRGN. However, serum anti-FB antibodies were not re-evaluated following the resolution of the IE, and it is possible they normalized.

PR3-ANCA was also positive in our patient. There have been some reports of infection-induced ANCA-associated vasculitis (AAV). One case report showed that granulomatosis with polyangiitis can be infection-induced [14]. In contrast, PR3-ANCA may appear in association with non-AAV diseases, such as infection, sarcoidosis, and ulcerous colitis [15]. No evidence of AAV in the kidney biopsy was found concerning our patient, nor did we observe any signs or symptoms of AAV. We consider that PR3-ANCA was induced by IE and that it was not pathogenic.

In conclusion, we report the first case of infective endocarditis-induced cryoglobulinemic glomerulonephritis. The association between glomerulonephritis and infection was shown using two different markers for IRGN (NAPlr/plasmin staining and serum anti-FB antibodies). Pathogens other than hepatitis C may possibly cause cryoglobulinemic glomerulonephritis. Moreover, in cases of infection-associated glomerulonephritis, factors such as IRGN, AAV, and cryoglobulinemic glomerulonephritis should be considered in relation to etiology. 

## Figures and Tables

**Figure 1 ijms-24-09369-f001:**
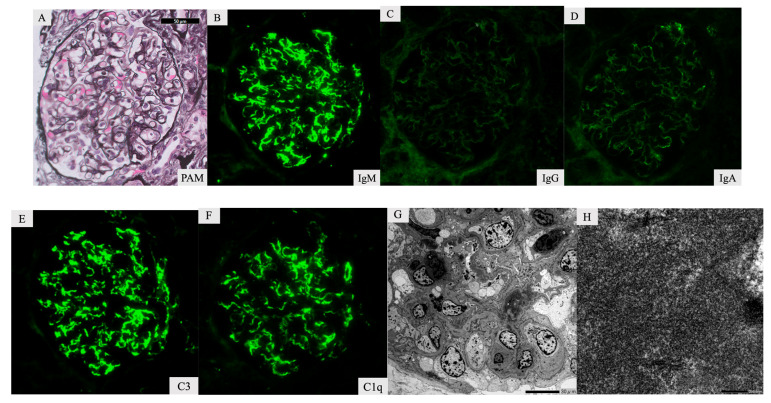
Kidney biopsy specimen. Light microscopy shows slight endocapillary proliferation with neutrophils and lymphocytes (**A**). Immunofluorescence staining showed that IgM (**B**), C3 (**E**), and C1q (**F**) were strongly positive in the ciliary loop. Immunoglobulin G (IgG), (**C**), and IgA (**D**) tests were negative. Electron microscopy showed high electron density deposits in the mesangial area and subendothelial (**G**). In a high-power field, the deposits exhibited a fibrous structure (**H**).

**Figure 2 ijms-24-09369-f002:**
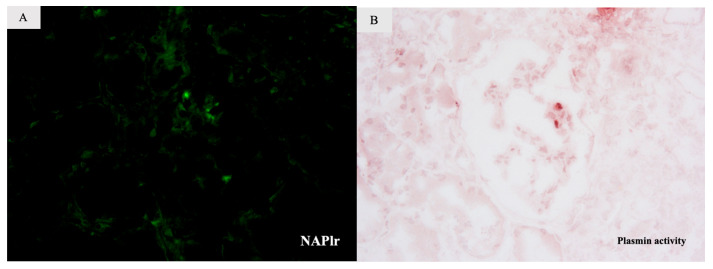
Additional tests of a kidney biopsy specimen. Both NAPlr (**A**) and plasmin activity (**B**) were segmentally and weakly positive in glomeruli of sequential sections of fresh frozen tissue. The distribution of the positive portion for those staining was similar (**A**,**B**).

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
