# Peer review of "Infection-Related Cryoglobulinemic Glomerulonephritis with Serum Anti-Factor B Antibodies Identified and Staining for NAPlr/Plasmin Activity Due to Infective Endocarditis"

_ijms, 2023, doi:10.3390/ijms24119369_

Round 1
Reviewer 1 Report
AT row 95 interstitial "lesion": it sounds better parenchyma
at row 107 "were strongly positive in the ciliary loop": the authors mean capillary?
at row 108 ". Electron microscopy showed high electron density deposits in the mesangial area (1G)." It seems to me odd that with a prevalent capillary pattern in immunofluorescence and with a proliferative endocapillary pattern on light microscopy the electron microscopy pattern is mesangial and truly speaking the deposits seem to me not only mesangial but also subendothelial.
Author Response
Thank you for your review.
I changed "lesion" at row 95 to "parenchyma".
"strongly positive in the ciliary loop" at row 107 means capillary.
As you pointed out at row 108 , the deposits exist not only mesangial but also subendothelial. I added the word "subendothelial".
Reviewer 2 Report
Dear Authors,
I read with interest your paper. I think the case is well-described and written. I only suggest adding some additional technical information about the methods to detect tissue NAPlr deposition and plasmin activity. I think it could be useful and interesting for the readers.
Author Response
Thank you for your review.
I added the methods to detect tissue NAplr deposition and plasmin activity.
Please check the manuscript.
kinds regard.
Reviewer 3 Report
Dear authors,
thank you for this excellent case presentation. Cases of cryoglobulinemic GN associated with infection have been described. That makes your case rare but not the first one. Please describe adequately what makes this particular case unique. For example stress out the role of NAPlr/Plasmin activity, and serum anti-FB antibodies as markers of infection-associated GN.
All of my best regards.
Author Response
Thank you for your review.
As you pointed out, I removed the sentence " This is the first reported association between cryoglobulinemic glomerulonephritis and bacterial infection".
kinds regard.